# Encapsulation of β-Galactosidase into Polyallylamine/Polystyrene Sulphonate Polyelectrolyte Microcapsules

**DOI:** 10.3390/ijms252010978

**Published:** 2024-10-12

**Authors:** Yuri S. Chebykin, Egor V. Musin, Aleksandr L. Kim, Sergey A. Tikhonenko

**Affiliations:** Institute of Theoretical and Experimental Biophysics Russian Academy of Science, Institutskaya St., 3, 142290 Puschino, Moscow Region, Russia; kobepoftruth@gmail.com (Y.S.C.); eglork@gmail.com (E.V.M.); kimerzent@gmail.com (A.L.K.)

**Keywords:** encapsulation, β-galactosidase, polyelectrolyte microcapsules, microcapsules, PMC, lactase, lactose, hypolactasia, polyallylamine, polystyrene sulphonate

## Abstract

More than half of the global population is unable to consume dairy products due to lactose intolerance (hypolactasia). Current enzyme replacement therapy methods are insufficiently effective as a therapeutic approach to treating lactose intolerance. The encapsulation of β-galactosidase in polyelectrolyte microcapsules by using the layer-by-layer method could be a possible solution to this problem. In this study, adsorption and co-precipitation methods were employed for encapsulating β-galactosidase in polyelectrolyte microcapsules composed of (polyallylamine /polystyrene sulphonate)₃. As a result, the co-precipitation method was chosen for β-galactosidase encapsulation. The adsorption method permits to encapsulate six times less enzyme compared with the co-precipitation method; the β-galactosidase encapsulated via the co-precipitation method released no more than 20% of the initially encapsulated enzyme in pH 2 or 1 M NaCl solutions. In contrast, when using the sorption method, about 100% of the initially encapsulated enzyme was released from the microcapsules under the conditions described above. The co-precipitation method effectively prevents the complete loss of enzyme activity after 2 h of incubation in a solution with pH 2 while also alleviating the adverse effects of ionic strength. Consequently, the encapsulated form of β-galactosidase shows promise as a potential therapeutic agent for enzyme replacement therapy in the treatment of hypolactasia.

## 1. Introduction

Milk and dairy products serve as rich sources of protein and essential micronutrients for over 6 billion people worldwide [1,2,3,4,5,6,7]. However, despite their easily digestible form, more than half of the global population is unable to consume dairy products due to lactose intolerance (hypolactasia) [8]. This condition is associated with insufficient secretion of the active form of β-galactosidase (lactase) in the small intestine [9], which normally hydrolyzes disaccharide lactose at the β(1 → 4) bond into its monomers, galactose and glucose [10]. Lactose is one of the primary carbohydrates in dairy products and is also widely used in pharmaceutical production [11,12]. Therefore, the inability to consume these products in individuals with hypolactasia significantly decreases their quality of life [13,14] and necessitates the regular use of medication.

The exclusion of lactose-containing products from the diet is one of the most common approaches to addressing this issue. However, long-term avoidance of dairy products without compensatory therapy can lead to the development of several diseases, such as osteoporosis, osteomalacia, and hypertension [15,16]. The use of prebiotics and probiotics, such as *Bifidobacterium* [17] or *Lactobacillus* [18], to enhance lactose metabolism by intestinal bacteria offers a potential solution to the aforementioned challenges [19]. However, this approach may not be suitable for everyone, as it is effective only in cases of hypolactasia caused exclusively by the absence of these specific microbiota in the intestine.

Enzyme replacement therapy is a more effective therapeutic approach to treating lactose intolerance. The use of β-galactosidase (lactase) derived from non-human sources [20,21] to hydrolyze lactose in whole milk or through oral administration addresses the issue in individuals with hypolactasia. However, enzyme inactivation due to low gastric pH and proteolytic enzymes in the gastrointestinal tract significantly reduces its efficacy, resulting in the short-term effect of this therapy [22]. Consequently, the development of immobilized forms of lactase for enzyme replacement therapy has gained attention. However, these forms also have several limitations: enzyme release in the small intestine leads to a gradual decrease in concentration over time [23,24], and the immobilizing matrix reduces the activity of other enzymes and impairs glucose absorption [25,26,27,28] or induces intestinal inflammation [29]. Given these challenges, there is a need to develop new immobilized forms of lactase for enzyme replacement therapy.

The encapsulation of β-galactosidase (lactase) in polyelectrolyte microcapsules may offer a solution to the aforementioned challenges. Polyelectrolyte microcapsules are produced by the sequential adsorption of oppositely charged polyelectrolytes onto a microparticle, which is subsequently removed in the final step [30,31]. The shell of the polyelectrolyte microcapsules is semipermeable, meaning it is permeable to low-molecular-weight compounds but impermeable to high-molecular-weight substances (up to 70 kDa [32]). This property allows the substrate to access the enzyme and the reaction products, galactose and glucose, to be released into the incubation medium, while the enzyme remains confined within the polyelectrolyte microcapsule. Notably, previous studies have demonstrated the successful encapsulation of enzymes within polyelectrolyte microcapsules with retention of their activity, including glucose oxidase [33,34,35], peroxidase [35], urease [36,37], lactate dehydrogenase [38], and alcohol dehydrogenase [39]. However, to date, no studies have focused on the encapsulation of β-galactosidase in polyelectrolyte microcapsules. β-Galactosidase was also encapsulated into polyelectrolyte microcapsules, as demonstrated in the study by Meenakshi Gupta et al. [40]. The aim of the study was to treat GM1 gangliosidosis, a lysosomal storage disorder, through the intracellular delivery of encapsulated β-galactosidase. The authors encapsulated approximately 30 mU per capsule by using dextran sulfate and poly-L-arginine on β-gal-loaded CaCO_3_ cores, showing activity in three cell types. However, such capsules are unsuitable for addressing the limitations of enzyme replacement therapy described above. Dextran sulfate and poly-L-arginine capsules are degradable by proteases and acidic gastric conditions, which would prevent the protection of encapsulated β-galactosidase under these conditions, hindering the desired therapeutic effect. Therefore, a different polyelectrolyte composition for the microcapsule shell is needed. In this work, we propose the use of polystyrene sulfonate and polyallylamine, as these polyelectrolytes are resistant to proteolytic degradation and low pH. Moreover, microcapsules made from these polyelectrolytes are well studied, including for enzyme encapsulation [41].

The aim of this study is to investigate the possibility of encapsulating β-galactosidase in polyelectrolyte microcapsules while preserving the enzyme’s activity. A secondary aim is to examine the effects of ionic strength and pH values, similar to those found in the gastric and intestinal environments, on the activity of encapsulated β-galactosidase. This will enable the future assessment of the potential use of this technology in enzyme replacement therapy.

## 2. Results and Discussion

To investigate the possibility of encapsulating β-galactosidase (lactase) in polyelectrolyte microcapsules (PMCs) while preserving its activity, two enzyme encapsulation methods were employed: sorption and co-precipitation. The encapsulation schemes of β-galactosidase in polyelectrolyte microcapsules using these methods are presented in Figure 1.

The sorption method (Figure 1A) involves the preparation of polyelectrolyte microcapsules followed by their incubation in an enzyme solution. In the first step, polyallylamine and polystyrene sulfonate are alternately adsorbed onto CaCO_3_ particles, which are subsequently dissolved by using ethylenediaminetetraacetic acid (EDTA). As a result, polyelectrolyte microcapsules (PMCs) with a complex internal nanocomposite structure are obtained [42,43]. Then, PMCs are incubated in a β-galactosidase solution to allow for enzyme adsorption. The main feature distinguishing the co-precipitation method from the sorption method lies in the fact that during co-precipitation, β-galactosidase is added directly to the solution during the formation of CaCO_3_ particles (Figure 1B). This results in the CaCO_3_ particles capturing β-galactosidase molecules and co-precipitating with them. Therefore, the final step of “enzyme adsorption” is not required for enzyme encapsulation. The presence of protein in the composition of CaCO_3_ particles results in the PMCs obtained from them having a distinct shell structure [38,41], which prevents the enzyme molecules from leaking into the incubation medium, while still allowing the substrate and reaction products to pass through.

As a result, β-galactosidase was encapsulated by using the methods described above. The obtained results are presented in Figure 2.

As shown in Figure 2A, the co-precipitation method encapsulates approximately 1300 Units (U) of the enzyme in 5 × 10^7^ polyelectrolyte microcapsules, whereas the sorption method allows for the adsorption of only 200 U. Additionally, Figure 2B demonstrates that maximum enzyme adsorption in 10^8^ microcapsules is observed after 60 min of incubation. Therefore, it can be concluded that the co-precipitation method enables the encapsulation of six times more enzyme units.

In the next stage of this research study, the release of the enzyme over time from polyelectrolyte microcapsules of both types was studied in relation to the pH of the solution. This stage is essential, as the encapsulated form of β-galactosidase is expected to overcome the acidic environment of the stomach, where the pH typically ranges from 1.5 to 3.5 [44,45,46,47]. After passing through the acidic conditions of the stomach, the polyelectrolyte microcapsules (PMCs) are exposed to pancreatic juice, which has a pH ranging from 7.5 to 8.5 [48,49]. Therefore, the PMCs were also incubated at pH 8. The results of β-galactosidase release from microcapsules prepared by using the sorption (PMCsorb) and co-precipitation (PMCcop) methods are presented in Figure 3.

As shown in Figure 3A, the amount of β-galactosidase released from PMCcop after 120 h of incubation does not exceed 20% of the amount initially encapsulated. Lowering the pH of the solution results in a reduction in the interaction between the polyelectrolytes, leading to the rearrangement of the polyelectrolyte complex and the possible formation of pores in the shell through which the encapsulated protein is released into the incubation medium [41,50,51]. At pH 6 and 8, enzyme release is observed to be no more than 6%. In contrast, for PMCsorb, 100% release of the enzyme from the capsules is observed at pH 2, while at pH 6 and 8, enzyme release does not exceed 72%. This effect is presumably related to the pronounced shell of PMCcop, which is impermeable to the enzyme. In the case of PMCsorb, the microcapsules lack a distinct shell capable of preventing β-galactosidase release. Additionally, a decrease in the pH of the medium leads to the protonation of polyallylamine, reducing its interaction with the negatively charged regions of β-galactosidase [52], consequently resulting in enzyme release.

In the next phase of the study, the release of the enzyme from polyelectrolyte microcapsules was investigated over time as a function of the ionic strength of the solution. This phase is crucial, as the encapsulated form of β-galactosidase is intended for use in replacement therapy in the small intestine, where the ionic strength of the environment may reach up to 0.5 M NaCl [53,54,55]. The results of the release of β-galactosidase, encapsulated by using the sorption (PMCsorb) and co-precipitation (PMCcop) methods, are presented in Figure 4.

As seen in Figure 4A, the amount of β-galactosidase released from PMCcop does not exceed 17% of the amount initially encapsulated, with no significant variation in enzyme release behavior observed with changes in NaCl concentration. In contrast, for PMCsorb, there is substantial release of the enzyme from the capsules, which can reach up to 76% of the initially adsorbed amount at 0.5 M NaCl. Additionally, Figure 4B shows that increasing the ionic strength of the solution to 1 M NaCl leads to nearly complete release of β-galactosidase from PMCsorb (almost 100%). Given the electrostatic nature of enzyme adsorption, an increase in ionic strength results in screening effects between the protein and the polyelectrolyte of the capsule, leading to the release of the substance [39]. Therefore, based on the aforementioned experiments, it can be concluded that the co-precipitation method is more suitable for encapsulating β-galactosidase.

Since the co-precipitation method proves to be more suitable for encapsulating β-galactosidase, it was proposed to investigate its catalytic properties. Firstly, the enzyme activity and Km of the encapsulated β-galactosidase were determined. Enzyme activity assesses whether the functionality of the enzyme is retained after encapsulation, while Km evaluates changes in the enzyme’s affinity for the substrate. The results are presented in Figure 5.

Figure 5A shows that the encapsulated form of β-galactosidase retains 10% of the initial enzyme activity (with the enzyme quantity being the same). This reduction in activity is likely due to the inhibition of the enzyme by the polyelectrolytes that constitute the microcapsule shell, as observed with urease and alcohol dehydrogenase [38,39]. To examine Km, the amount of polyelectrolyte microcapsules containing β-galactosidase was increased (10 times more enzyme units than in the free enzyme) to better illustrate the enzyme’s activity dependence on lactose concentration. As shown in Figure 5B, the Km of the encapsulated β-galactosidase was 96 mM, compared with a Km of 36 mM for the free enzyme. Given that Km increased by 2.7 times and activity decreased by 90%, it is likely that a mixed type of enzyme inhibition occurred. Nevertheless, despite the significant reduction in enzyme activity and substrate affinity, this encapsulated form of β-galactosidase could still be used in enzyme replacement therapy in the future, as the change in Km is not substantial and the reduced activity could be compensated by using a larger quantity of microcapsules containing the enzyme.

In the subsequent stage, the influence of the ionic strength of the solution on the activity of encapsulated β-galactosidase was investigated. Similarly to the aforementioned study, this stage is essential, as the ionic strength of the small-intestine environment can reach up to 0.5 M NaCl [53,54,55], and the ionic strength of the solution, in turn, may affect enzyme activity. The results obtained are presented in Figure 6.

Figure 6A shows that the activity of the encapsulated enzyme remains unchanged regardless of the ionic strength of the solution, whereas the activity of the native enzyme decreases with the increase in NaCl concentration, with a 3.5-fold reduction observed at 1 M NaCl. Prolonged incubation of encapsulated β-galactosidase in a 1 M NaCl solution also demonstrates the preservation of enzyme activity without any noticeable changes.

In the next stage, the effect of solution pH on the activity of encapsulated β-galactosidase was investigated. This study is necessary for the same reason outlined above: it is anticipated that the microcapsules can withstand the acidic environment of the stomach and hydrolyze lactose in the small intestine. Consequently, encapsulated β-galactosidase was incubated for a specified period, after which the capsules were removed from the medium and their activity was measured at pH 6, which is characteristic of the small intestine’s acidity. Additionally, according to the literature, the pH of the small intestine can briefly reach 8 due to the neutralization of gastric juice with sodium phosphate [48,49]. Therefore, changes in the activity of the encapsulated enzyme after brief incubation at pH 8 were also investigated. The results obtained are presented in Figure 7.

As shown in Figure 7A, the activity of both encapsulated and native β-galactosidase remains relatively stable after brief incubation in solutions with pH 2 and 8. However, Figure 7B reveals that with prolonged incubation of encapsulated β-galactosidase at pH 2, there is a gradual decrease in enzyme activity over a period of 2 h. Specifically, after 2 h of incubation at pH 2, the activity of the encapsulated enzyme decreases by a factor of 5.5. Nonetheless, the PMC coating stabilized the enzyme, as the native enzyme lost its activity within 20 min. This stabilization allows the enzyme to maintain its activity for the necessary period required for gastric emptying.

As a result of this work, the co-precipitation method was selected for the encapsulation of β-galactosidase. This method effectively prevents the complete loss of enzyme activity after 2 h of incubation in a solution with pH 2, while also mitigating the negative effects of ionic strength. Consequently, the obtained form of encapsulated β-galactosidase holds promise as a potential therapeutic agent in enzyme replacement therapy for the treatment of hypolactasia. Furthermore, the modular structure of the preparation of polyelectrolyte microcapsules allows for future modifications aimed at reducing enzyme activity loss due to encapsulation and exposure to the acidic environment of the stomach.

## 3. Materials and Methods

### 3.1. Materials

Polyelectrolytes sodium polystyrene sulfonate (PSS) and polyallylamine hydrochloride (PAH) with a molecular weight of 70 kDa were purchased from Merch (St. Louis, MS, USA) (residual monomer < 10%). Glucose oxidase (150 U/mg) was purchased from Merch (St. Louis, MS, USA). β-Galactosidase was purchased from HUANWEI BIOTECH (Shijiazhuang, Hebei Province, China). Lactose, sodium chloride, sodium hydroxide, and hydrochloric acid were from “Reakhim” (Moscow, Russia). The percentage content of each chemical is at least 99.9%.

### 3.2. Encapsulation of Enzymes in Polyelectrolyte Microcapsules (Co-Precipitation Method)

To prepare 10 million polyelectrolyte microcapsules, calcium carbonate (CaCO_3_) microspherulites (average size of 4.5 ± 1 µm) were first synthesized. For this, 1.5 mL of 0.33 M Na_2_CO_3_ was added to 1.5 mL of a 0.33 M CaCl_2_ solution containing 6 mg/mL of the enzyme, under intense stirring [30]. The resulting suspension was centrifuged at 500 g, and the supernatant was decanted before using the pellet for the preparation of polyelectrolyte microcapsules (PMCs). The polyelectrolyte shell was formed by alternately incubating the CaCO_3_ microspherulites in solutions of the polyanion poly(styrene sulfonate) (PSS) and the polycation poly(allylamine hydrochloride) (PAH), both at concentrations of 2 mg/mL in 2 mL volumes containing 0.5 M NaCl. After each incubation, samples were washed three times with 2 mL of 0.5 M NaCl to remove non-adsorbed polymer molecules. After applying the necessary number of layers, the calcium carbonate core was dissolved and removed by incubating the microcapsules in 20 mL of 0.2 M EDTA solution (pH 7) for 2 h. In the final step, the resulting polyelectrolyte microcapsules were washed three times with 2 mL of distilled water. The obtained PMCs had a composition of (PAH/PSS)_3_, where PAH acts as the first layer and PSS as the last layer in PMC formation, and three is the number of PAH/PSS bilayers. The average size of the obtained PMCs was 4.5 ± 1 µm. The size and number of microparticles were measured by using dynamic light scattering with a Zetasizer nano ZS (Malvern, UK).

### 3.3. Encapsulation of Enzymes in Polyelectrolyte Microcapsules (Adsorption Method)

This method is similar to the co-precipitation method, with the key distinction being that for the creation of these polyelectrolyte microcapsules, calcium carbonate (CaCO_3_) microspherulites without the enzyme were used as the core [56]. After forming the polyelectrolyte shell, the CaCO_3_ core was dissolved and removed by incubation in 20 mL of 0.2 M EDTA solution for 2 h. The resulting polyelectrolyte microcapsules were washed three times with 2 mL of distilled water. Subsequently, the microcapsules were incubated in a protein solution with a concentration of 6 mg/mL for 12 h. The microcapsules were washed three times with bidistilled water to remove non-adsorbed protein molecules. The obtained PMCs had a composition of (PAH/PSS)_3_, where PAH acts as the first layer and PSS as the last layer in PMC formation, and three is the number of PAH/PSS bilayers. The average size of the resulting polyelectrolyte microcapsules was 4.5 ± 1 µm. The size and number of microparticles were measured by using dynamic light scattering with a Zetasizer nano ZS (Malvern, UK).

### 3.4. Measurement of β-Galactosidase Activity

Enzyme activity is estimated by the number of substrate molecules that are converted into product molecules in a definite time [57]. To measure β-galactosidase activity, a sequential reaction of β-galactosidase and glucose oxidase was used. Glucose oxidase was added in excess to ensure that the enzymatic oxidation of glucose was not a rate-limiting step. Measurements were carried out by the change in dissolved oxygen concentration related to β-galactosidase activity added to the reaction cell. The rate of the enzymatic reaction was monitored by using a Clark oxygen electrode (model “Expert dk-1” (Econics Expert, Moscow, Russian)) in a 1 mL plexiglass chamber. The composition of the chamber for the sequential enzymatic reaction was as follows: glucose oxidase, 100 µg/mL (150 U/mg); lactose, 80 mg/mL; β-galactosidase, 40 Units, or PMCs containing encapsulated β-galactosidase in a quantity of 2.5 × 10^6^ microcapsules. Unless otherwise stated in the text, the standard conditions under which the reaction was conducted were pH 6 and 24 °C (room temperature).

### 3.5. Determination of Km of β-Galactosidase

To determine the Michaelis constant (Km), the activity of free and encapsulated β-galactosidase was measured as a function of the lactose concentration in the sample. The Km values were determined as the concentration of substrate at which the reaction rate is half of V_max_, which is the limiting rate approached by the system at saturating substrate concentration for a given enzyme concentration. The plotting of the graph and the determination of Km were performed according to the recommendations of the IUBMB (International Union of Biochemistry and Molecular Biology) [58].

### 3.6. Determination of β-Galactosidase Concentration

The amounts of encapsulated and released β-galactosidase were determined by using a spectrophotometric method. The absorbance of β-galactosidase at 280 nm was measured in the supernatant after CaCO_3_ precipitation (during the preparation of polyelectrolyte microcapsules by co-precipitation), in the supernatant after PMCs precipitation during the experiments to determine the amount of released enzyme, and in the supernatant after PMCs precipitation in the final stage of β-galactosidase encapsulation by adsorption. A UNICO 2804 spectrophotometer (Suite E, Dayton, NJ, USA) was used for the measurements.

### 3.7. Statistical Data Analysis

For each measurement of fluorescence intensity and enzyme activity, the mean values and relative standard deviations were calculated. The number of replicates (N) was 5. The significance of the differences was assessed by using an independent two-sample *t*-test (Student’s *t*-test), with *p* ≤ 0.01 being considered significant.

## 4. Conclusions

In this study, we investigated the possibility of encapsulating β-galactosidase in polyelectrolyte microcapsules and the effects of ionic strength and pH values, similar to those found in the gastric and intestinal environments, on the activity of encapsulated β-galactosidase.

As a result of the study, it was determined that the co-precipitation method is the most suitable for encapsulating β-galactosidase in polyelectrolyte microcapsules compared with the adsorption method. This conclusion is based on the following observations: the adsorption method permits to encapsulate six times less enzyme compared with the co-precipitation method; the β-galactosidase encapsulated via adsorption is fully released from the microcapsules at pH 2; and the enzyme encapsulated via adsorption is entirely released from the microcapsules in the presence of an ionic strength of 1 M NaCl. In contrast, when using the co-precipitation method, no more than 20% of the initially encapsulated enzyme is released from the microcapsules under the conditions described above.

Subsequently, the catalytic properties of both the native β-galactosidase and the β-galactosidase encapsulated in the polyelectrolyte microcapsules were investigated. The encapsulated form of β-galactosidase retains 10% of the initial enzyme activity (with the enzyme quantity being the same). The Km of the encapsulated β-galactosidase is 96 mM, compared with a Km of 36 mM for the free enzyme. Nevertheless, despite the significant reduction in enzyme activity and substrate affinity, this encapsulated form of β-galactosidase could still be used in enzyme replacement therapy in the future, as the change in Km is not substantial and the reduced activity could be compensated by using a larger quantity of microcapsules containing the enzyme. Moreover, the encapsulated form of the enzyme demonstrated a significant advantage over the native enzyme form for potential use in the treatment of hypolactasia. It was shown that ionic strength of up to 1 M NaCl did not affect the activity of the encapsulated enzyme, whereas the activity of the native enzyme decreased by a factor of 3.5. Additionally, it was observed that at pH 2, the native enzyme completely lost its activity after 20 min of incubation, whereas the activity of the encapsulated enzyme decreased by a factor of 5.5 after 2 h of incubation at pH 2. This stabilization allows the enzyme to maintain its activity for the necessary period required for gastric emptying.

The encapsulated β-galactosidase form shows potential as a therapeutic agent for enzyme replacement therapy in treating hypolactasia. Additionally, the modular design of the polyelectrolyte microcapsules offers opportunities for future adjustments to minimize enzyme activity loss during encapsulation and when exposed to the stomach’s acidic environment.

## Figures and Tables

**Figure 1 ijms-25-10978-f001:**
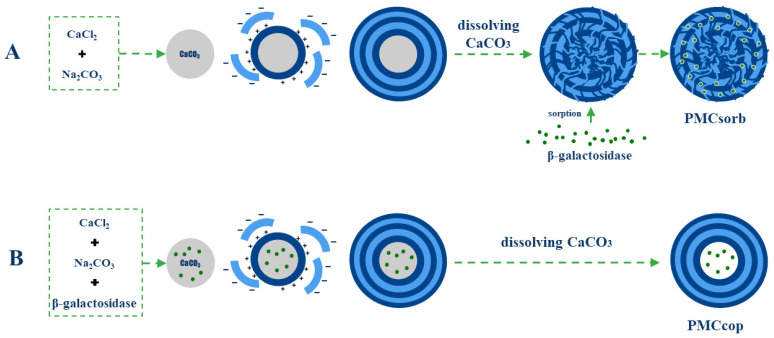
Schemes of β-galactosidase encapsulation in polyelectrolyte microcapsules using the sorption method (**A**) and the co-precipitation method (**B**).

**Figure 2 ijms-25-10978-f002:**
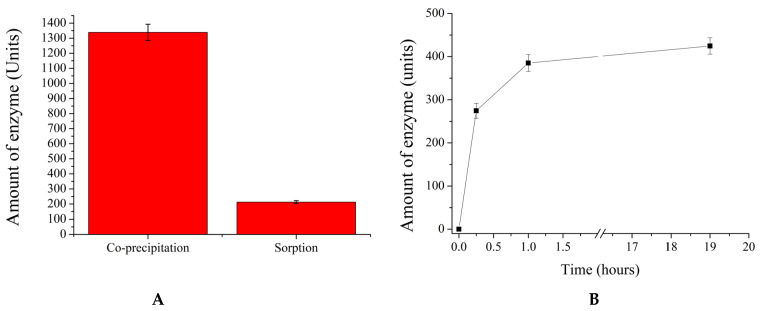
Amounts of β-galactosidase encapsulated by the sorption and co-precipitation methods (**A**). Sorption dynamics of β-galactosidase encapsulated by the sorption method (**B**).

**Figure 3 ijms-25-10978-f003:**
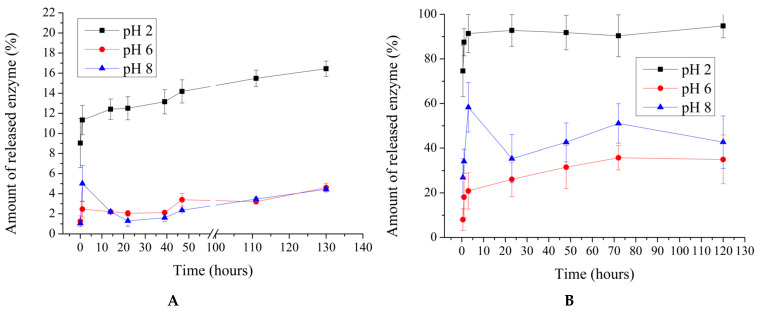
Release of β-galactosidase from polyelectrolyte microcapsules as a function of pH. Co-precipitation method (**A**); sorption method (**B**).

**Figure 4 ijms-25-10978-f004:**
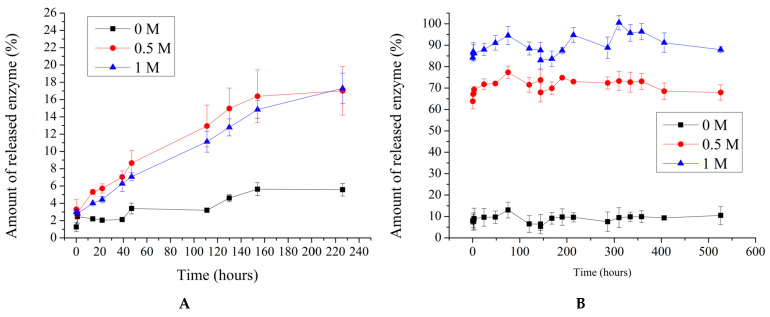
Release of β-galactosidase from polyelectrolyte microcapsules as a function of NaCl concentration. Co-precipitation method (**A**); adsorption method (**B**).

**Figure 5 ijms-25-10978-f005:**
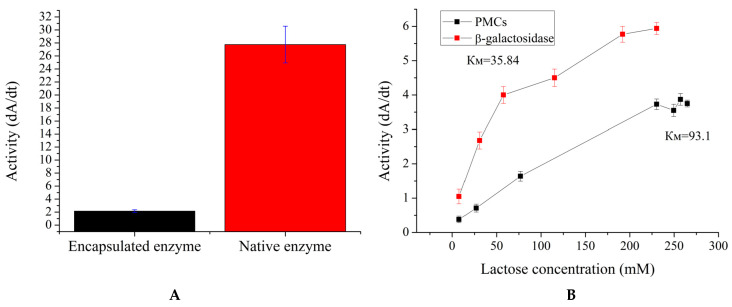
Activity of encapsulated and free β-galactosidase (**A**). Km of free and encapsulated β-galactosidase (**B**).

**Figure 6 ijms-25-10978-f006:**
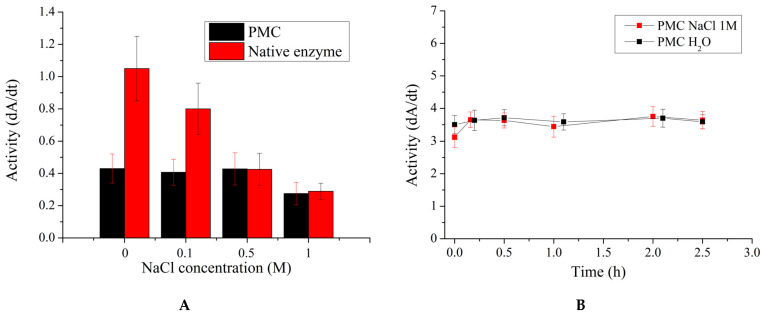
Activity of encapsulated and free β-galactosidase after incubation in solutions with varying ionic strengths (**A**). Variation in activity of encapsulated β-galactosidase upon incubation in 1 M NaCl solution (**B**).

**Figure 7 ijms-25-10978-f007:**
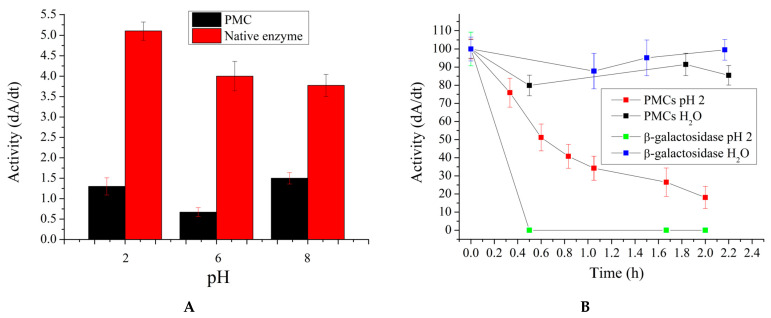
Activity of encapsulated and native β-galactosidase after incubation in solutions with varying pH values (**A**). Variation in activity of β-galactosidase following incubation in solutions with pH 2 (**B**).

## Data Availability

Data is contained within the article.

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
