# Peer review of "Encapsulation of β-Galactosidase into Polyallylamine/Polystyrene Sulphonate Polyelectrolyte Microcapsules"

_ijms, 2024, doi:10.3390/ijms252010978_

Round 1

Reviewer 1 Report

Comments and Suggestions for Authors

This is an interesting paper describing the results of the encapsulation of beta-galactosidase into polyelectrolyte microcapsules. It is recommended to accept this manuscript for publication after some revision on the basis of comments below.

COMMENTS

1.

The title is not specific sufficiently. It should specifically mention what kind of polyelectrolytes are used. Therefore, the following title is suggested:

Encapsulation of beta-galactosidase into polyallylamine/polystyrene sulphonate polyelectrolyte microcapsules

2.

In the manuscript text, either in the Introduction or in the Results and Discussion section, the authors have to explain why polyallylamine/polystyrene sulphonate polyelectrolytes were selected by them, and not other well-known polyelectrolytes.

3.

In line 15, there is a small number of 3 in subscript. What is the meaning of this number?

4.

The authors have found that the activity of the examined beta-galactosidase is significantly lower in the case of microcapsules obtained by the sorption method than that in the case of coprecipitation. Why? The authors should provide explanation.

5.

In Figure 2, the “Amount of enzyme (Units)” is written on the y-axis. However, they do not provide any definition what they exactly mean on “unit”. This should be provided.

6.

In line 156, correctly Figure 4B (and not 3B).

7.

The definition of activity is not provided in either the Experimental or in the Results and Discussion sections. This should be given.

8.

In line 185, “… for by …” is written. Correctly only “… by …”.

9.

The authors should describe in details how they determined the Km. There is no any Michaelis-Menten kinetic evaluation presented. The authors should present the details of such measurements and the kinetic plots for the Km determination.

10.

In Figure 7A, it is not provided what the authors mean to present on the x-axis. pH?

11.

In line 274, the quantity of microcapsules is given. How this value was determined. No any experimental details are provided. Therefore, interested scientists and engineers will not be able to reproduce such data. Reproducibility is the most important requirement for a scientific work and publication.

Author Response

Comment 1. 

The title is not specific sufficiently. It should specifically mention what kind of polyelectrolytes are used. Therefore, the following title is suggested:

Encapsulation of beta-galactosidase into polyallylamine/polystyrene sulphonate polyelectrolyte microcapsules

Response 1: 

Corrections have been made.

Comment 2. 

In the manuscript text, either in the Introduction or in the Results and Discussion section, the authors have to explain why polyallylamine/polystyrene sulphonate polyelectrolytes were selected by them, and not other well-known polyelectrolytes.

Response 2: 

An explanation of the choice of these polyelectrolytes has been added to the Introduction: 

" β-galactosidase was also encapsulated into polyelectrolyte microcapsules, as demon-strated in the study by Meenakshi Gupta et al [40]. The aim of this study is to treat GM1 gangliosidosis, a lysosomal storage disorder, through the intracellular delivery of encapsulated β-galactosidase. The authors encapsulated approximately 30 mU per capsule using dextran sulfate and poly-L-arginine on β-gal-loaded CaCO3 cores, showing activity in three cell types. However, such capsules are unsuitable for ad-dressing the limitations of enzyme replacement therapy described above. Dextran sul-fate and poly-L-arginine capsules are degradable by proteases and acidic gastric con-ditions, which would prevent the protection of encapsulated β-galactosidase under these conditions, hindering the desired therapeutic effect. Therefore, a different poly-electrolyte composition for the microcapsule shell is needed. In this work, we propose the use of polystyrene sulfonate and polyallylamine, as these polyelectrolytes are re-sistant to proteolytic degradation and low pH. Moreover, microcapsules made from these polyelectrolytes are well-studied, including for enzyme encapsulation [41]."

Comment 3. 

In line 15, there is a small number of 3 in subscript. What is the meaning of this number?

Response 3: 

The number "3" refers to the number of polyelectrolyte layer pairs (polyallylamine/polystyrene sulphonate), where polyallylamine forms the first layer and polystyrene sulphonate the last. In total, there are six layers: PAH/PSS/PAH/PSS/PAH/PSS. This notation has been clarified in the manuscript.

Comment 4. 

The authors have found that the activity of the examined beta-galactosidase is significantly lower in the case of microcapsules obtained by the sorption method than that in the case of coprecipitation. Why? The authors should provide explanation.

Response 4: 

We did not study the activity of beta-galactosidase encapsulated by the sorption method. We avoided investigating this due to the acidic pH and ionic strength (greater than 0.5 M NaCl) causing protein release from sorption-derived capsules. This does not meet our goal of encapsulating beta-galactosidase for prolonged functionality. 

Comment 5. 

In Figure 2, the “Amount of enzyme (Units)” is written on the y-axis. However, they do not provide any definition what they exactly mean on “unit”. This should be provided.

Response 5: 

The enzyme unit, or international unit for enzyme (symbol U, sometimes IU), is a measure of catalytic activity. One unit (1 U, or μmol/min) is defined as the amount of enzyme that catalyzes the conversion of one micromole of substrate per minute under the specified conditions of the assay method. This unit was adopted by the International Union of Biochemistry in 1964. 

Since this is a universally accepted unit of measurement, it does not require further clarification according to the journal guidelines (IJMS).

Comment 6. 

In line 156, correctly Figure 4B (and not 3B).

Response 6: 

The authors thank the reviewer. The correction has been made.

Comment 7. 

The definition of activity is not provided in either the Experimental or in the Results and Discussion sections. This should be given.

Response 7: 

The definition of activity has been added to the Experimental section.

Comment 8. 

In line 185, “… for by …” is written. Correctly only “… by …”.

Response 8: 

Corrections have been made.

Comment 9. 

The authors should describe in detail how they determined the Michaelis constant (Km). There is no Michaelis-Menten kinetic analysis presented. The authors should include these measurements and provide the kinetic plots for Km determination.

Response 9: 

A new section detailing the Km determination method has been added to the Materials and Methods section.

Comment 10. 

In Figure 7A, it is not provided what the authors mean to present on the x-axis. pH?

Response 10: 

Yes, the x-axis represents pH values. Corrections have been made.

Comment 11. 

In line 274, the quantity of microcapsules is given. How this value was determined. No any experimental details are provided. Therefore, interested scientists and engineers will not be able to reproduce such data. Reproducibility is the most important requirement for a scientific work and publication.

Response 11: 

The quantity of microcapsules was determined using a Zetasizer nano ZS (Malvern, UK). The model name has been added to the Materials and Methods section.

Reviewer 2 Report

Comments and Suggestions for Authors

1.     The author claimed “no studies have focused on the encapsulation of β-galactosidase in polyelectrolyte microcapsules”. However, a paper about encapsulation of β-galactosidase in polyelectrolyte microcapsules was not cited by the author. Please review this work and clarify the distinct contribution of their research. 

Gupta, M., Pandey, H., & Sivakumar, S. (2017). Intracellular delivery of β-galactosidase enzyme using arginase-responsive dextran sulfate/poly-l-arginine capsule for lysosomal storage disorder. ACS omega, 2(12), 9002-9012. https://pubs.acs.org/doi/full/10.1021/acsomega.7b01230

2.     What is the encapsulation efficacy (EE) of β-galactosidase

3.     Since β-galactosidase is being used as a drug in the body, ensuring its safety is crucial. Kindly provide evidence or references that support its safety profile. Alternatively, conduct cytotoxity assay to prove the biocompatibility. 

4.     In reference to Figure 6, could the author specify the pH conditions used during the experiments? 

Author Response

Comment 1.     The author claimed “no studies have focused on the encapsulation of β-galactosidase in polyelectrolyte microcapsules”. However, a paper about encapsulation of β-galactosidase in polyelectrolyte microcapsules was not cited by the author. Please review this work and clarify the distinct contribution of their research. 

Gupta, M., Pandey, H., & Sivakumar, S. (2017). Intracellular delivery of β-galactosidase enzyme using arginase-responsive dextran sulfate/poly-l-arginine capsule for lysosomal storage disorder. ACS omega, 2(12), 9002-9012. https://pubs.acs.org/doi/full/10.1021/acsomega.7b01230

Response 1

Response 1: 

We thank the reviewer for bringing this article to our attention. The description of the study has been added to the Introduction:

«β-galactosidase was also encapsulated into polyelectrolyte microcapsules, as demon-strated in the study by Meenakshi Gupta et al [40]. The aim of this study is to treat GM1 gangliosidosis, a lysosomal storage disorder, through the intracellular delivery of encapsulated β-galactosidase. The authors encapsulated approximately 30 mU per capsule using dextran sulfate and poly-L-arginine on β-gal-loaded CaCO3 cores, showing activity in three cell types. However, such capsules are unsuitable for ad-dressing the limitations of enzyme replacement therapy described above. Dextran sul-fate and poly-L-arginine capsules are degradable by proteases and acidic gastric con-ditions, which would prevent the protection of encapsulated β-galactosidase under these conditions, hindering the desired therapeutic effect. Therefore, a different poly-electrolyte composition for the microcapsule shell is needed. In this work, we propose the use of polystyrene sulfonate and polyallylamine, as these polyelectrolytes are re-sistant to proteolytic degradation and low pH. Moreover, microcapsules made from these polyelectrolytes are well-studied, including for enzyme encapsulation [41].»

However, this study does not diminish the novelty and relevance of our research:

  1. In the study by Meenakshi Gupta et al., the primary goal was to incorporate biodegradable (dextran sulfate /poly-l-arginine) capsules containing β-galactosidase into cells and demonstrate intracellular activity. The enzyme's activity was detected indirectly and only after capsule rupture and enzyme release, which does not meet the requirements for our purposes. These capsules are unsuitable for our goal, which is to provide prolonged stability for enzyme delivery through the gastrointestinal tract, where both stomach acid and intestinal proteases are present. Stomach acid and intestinal enzymes would degrade such capsules and the β-galactosidase, resulting in an outcome similar to administering free β-galactosidase. In contrast, we propose using PSS and PAH polyelectrolytes, which are resistant to these aggressive factors.

  1. Meenakshi Gupta et al. encapsulated the enzyme by adsorbing it onto preformed CaCO3 cores, followed by polyelectrolyte coating. In our work, we either co-precipitated the enzyme with CaCO3 during core formation, thus trapping it inside, or incubated preformed microcapsules in enzyme solution. The capsules produced by us and by Meenakshi Gupta et al. differ significantly in morphology and the method of enzyme immobilization. For instance, variations in pH and ionic strength lead to different distributions of protein in the capsules, resulting in different enzyme activities under such conditions and different durations of enzyme functionality.

  1. The study by Meenakshi Gupta et al. [40] also demonstrated the possibility of encapsulating β-galactosidase in PSS/PAH microcapsules. However, the authors used β-galactosidase-loaded 3-(aminopropyl)triethoxysilane (APTES)-functionalized silica templates as cores, which require removal with hydrofluoric acid (HF). This method is inappropriate because HF inactivates encapsulated biomolecules (10.1021/la036177z) and causes denaturation and complete degradation (Aimoto & Shimonishi, 1975; Edwards et al., 1984; Haugen & Suttie, 1974; Adamek et al., 2005).

  1. The authors used this type of PSS/PAH capsule only as a control in one experiment, where enzyme release was monitored over 12 hours in culture medium. These results do not allow for the conclusion that PSS/PAH microcapsules with β-galactosidase are viable for enzyme replacement therapy. In our research, we studied two methods of enzyme encapsulation into PSS/PAH microcapsules without using aggressive solvents or acids. We also examined enzyme release under conditions (pH and ionic strength) that simulate the stomach and small intestine environment and evaluated the catalytic properties of encapsulated β-galactosidase under both optimal and physiological conditions. None of these parameters were addressed in the study by Meenakshi Gupta et al.

Thus, the work by Meenakshi Gupta et al. has different goals and experimental outcomes, which do not overlap with our study, preserving the novelty and relevance of our research.

Comment 2: 

What is the encapsulation efficacy (EE) of β-galactosidase?

Response 2: 

The encapsulation efficiency of β-galactosidase is presented in Figure 2A and described in the caption: 

«As shown in Figure 2A, the co-precipitation method encapsulates approximately 1,300 Units (U) of the enzyme in 5*107 polyelectrolyte microcapsules, whereas the sorption method allows for the adsorption of only 200 U»

Comment 3: 

Since β-galactosidase is being used as a drug in the body, ensuring its safety is crucial. Kindly provide evidence or references that support its safety profile. Alternatively, conduct cytotoxity assay to prove the biocompatibility.

Response 3: 

In our study, β-galactosidase is encapsulated within a polyelectrolyte microcapsule, which prevents its direct contact with the external environment. Consequently, the authors fully agree with the reviewer that assessing the cytotoxicity of the proposed capsules is essential. There are studies demonstrating the absence of cytotoxicity of this type of capsule towards macrophages and cancer cells (10.1007/s10517-018-4291-7), C6 glioma and 3T3 fibroblast cells (https://doi.org/10.1016/j.colsurfb.2008.12.022), bacteriophages (10.3390/polym13060914; https://doi.org/10.3390/polym14030613), and bacteria (https://doi.org/10.3390/polym14030613). Therefore, there is reason to believe these microcapsules are likely safe. However, it is important to consider potential effects on the stomach/intestines, such as inflammation and whether the capsules will degrade over time in the intestines. This is an important issue, and we plan to investigate it in the future. However, this is beyond the scope of the current paper, as it is a large and time-consuming study requiring advanced equipment and funding. The data we published in this paper provide the foundation for future exploration of this project.

Comment 4: 

In reference to Figure 6, could the author specify the pH conditions used during the experiments?

Response 4: 

We thank the reviewer for this comment. The pH used was 6, and this information has been added to the Materials and Methods section.

Round 2

Reviewer 2 Report

Comments and Suggestions for Authors The manuscript has been improved to warrant publication in IJMS.